# Investigations of Single-Subunit tRNA Methyltransferases from Yeast

**DOI:** 10.3390/jof9101030

**Published:** 2023-10-19

**Authors:** Zhongyuan Wang, Xiangbin Xu, Xinhai Li, Jiaqi Fang, Zhenkuai Huang, Mengli Zhang, Jiameng Liu, Xiaoting Qiu

**Affiliations:** 1Ministry of Education Key Laboratory of Applied Marine Biotechnology, Ningbo University, Ningbo 315800, China; wzy245956@163.com (Z.W.); 15053937565@163.com (X.L.); fjq18758340317@163.com (J.F.); hzhenkuai@163.com (Z.H.); 18096770590@163.com (M.Z.); 15061566181@163.com (J.L.); 2College of Food and Pharmaceutical Sciences, Ningbo University, Ningbo 315800, China; chao_xuxiangbin@163.com; 3Li Dak Sum Yip Yio Chin Kenneth Li Marine Biopharmaceutical Research Centre, Ningbo University, Ningbo 315800, China

**Keywords:** tRNA methylation, yeast single-subunit TRM, Trm10p

## Abstract

tRNA methylations, including base modification and 2’-*O*-methylation of ribose moiety, play critical roles in the structural stabilization of tRNAs and the fidelity and efficiency of protein translation. These modifications are catalyzed by tRNA methyltransferases (TRMs). Some of the TRMs from yeast can fully function only by a single subunit. In this study, after performing the primary bioinformatic analyses, the progress of the studies of yeast single-subunit TRMs, as well as the studies of their homologues from yeast and other types of eukaryotes and the corresponding TRMs from other types of organisms was systematically reviewed, which will facilitate the understanding of the evolutionary origin of functional diversity of eukaryotic single-subunit TRM.

## 1. Introduction

Methylation is ubiquitous in DNA, RNA, proteins, lipids, polysaccharides, and small molecules in the cell [1,2,3]. tRNA is rich with diverse methylation positions. Due to the special location of modification, tRNA methylation plays an important role in tRNA structural stability, frameshift prevention, and translation efficiency [4]. Species in yeast, such as *Saccharomyces cerevisiae*, are model eukaryotic organisms with a high content of tRNA methylation, which is critical for the normal function of tRNA to ensure accuracy and efficiency during the process of protein biosynthesis [5,6]. With evolution, both the types and degrees of post-transcriptional modifications in tRNAs are increasing, suggesting that modified nucleotides in eukaryotic tRNAs have multiple important functions, which need to be further characterized [7]. 

Most tRNAs are composed of 73–94 nucleotides. Their secondary structure represents the shape of a cloverleaf, consisting of an amino acid acceptor stem, dihydrouracil loop (D loop), anticodon loop, thymine–pseudouridine–cytosine loop (TψC loop; T loop), variable loop, D stem, T stem, and anticodon stem; whereas the tertiary structure of tRNA is an inverted L-shape, in which the amino acid acceptor stem and T stem form the first coaxially stacked helical system and the D loop and anticodon loop form the second. This structure is maintained by intramolecular hydrogen bonds [8].

The exchange between cytoplasm and nucleus is a prerequisite for tRNA maturation including post-transcriptional modification [9]. Based on the function, the post-transcriptional modification of tRNA can be divided into two categories: (i) maintaining the structural stability of tRNA in vivo; (ii) affecting the efficiency and fidelity of translation [10,11,12]. From the view of the principle of physical chemistry, the modification of tRNA gives additional rigidity to the base or ribose moiety and provides a hydrophobic or hydrophilic environment for local structure, thus stabilizing the higher-order structure of tRNA [13,14]. Covalent modification of bases is an important prerequisite in the correct recognition of mRNA codons in ribosomes [15], and tRNA modification is especially critical for accurate and effective aminoacylation during decoding. The aminoacylation capability of tRNA is very poor in the absence of tRNA modification [16]. Furthermore, in order to maintain a high efficiency of translation, tRNA modifications are different among various tRNA species, thus standardizing the translation process [17].

The methylation modifications in tRNA include methylation of bases and ribose moiety (2′-*O*-methylation) as well as some more complicated chemical modifications. Many tRNA modifications are highly conserved in one or more kingdoms of life and frequently occur in the same position in tRNA. Methylation of the bases in tRNA represents one of the most abundant base modifications found in the three domains of life. For example, there are a large number of base methylations in human mitochondrial tRNAs, including m^1^A at positions 9, 16, and 58; m^5^U at position 54; m^3^C at position 32; m^5^C at positions 48, 49, and 50; m^1^G at positions 9 and 37; m^2^G and m^2,2^G at position 26 [18]. tRNA 2′-*O*-methylation modifications have also been discovered to be widely distributed in the three domains of life and have been identified at positions 4, 18, 32, 34, 39, 44, 54, and 56 (2′-*O*-methylations in positions 32 and 34 are ubiquitous) [19,20]. 

tRNA methylations in various positions are specifically catalyzed by different tRNA methyltransferases (TRMs), a group of post-transcriptional RNA-modifying enzymes responsible for transferring methyl groups from methyl donors (usually *S*-adenosyl-L-methionine (SAM)) to a variety of positions on the target RNA nucleotides [20]. For yeast multi-subunit TRMs, catalysis of the modification of substrate tRNA requires the efficient cooperation of multiple subunits. For example, depending on its partner subunit, Trm7p is capable of methylating positions 32 or 34 in substrate tRNA: a holoenzyme consisting of Trm7p and Trm732p is responsible for the generation of Cm32, while a holoenzyme consisting of Trm7p and Trm734p is responsible for the generation of Nm34 [21]; Trm8p and Trm82p jointly act on substrate tRNA to form m^7^G46 [22]; Trm11p and Trm112p are two subunits necessary for catalyzing the formation of m^2^G10 in substrate tRNA [23]. Another type of yeast TRM can play a complete catalytic role in modification in the form of only a single subunit. Here, we systematically review the research progress of yeast single-subunit TRMs in terms of modification positions, multiple functions, and structures, as well as the studies of homologues of these TRMs after performing the primary bioinformatic analyses, which provides a framework for further research on yeast single-subunit TRMs. As most of the yeast single-subunit TRMs possess homologues in humans and other higher eukaryotic species, studies of these TRMs will facilitate the realization of the evolutionary origin of functional diversity of eukaryotic single-subunit TRMs and thereby deepen our understanding of the mechanism of tRNA methylations.

## 2. tRNA Modifications Catalyzed by Yeast Single-Subunit TRMs

Some of the TRMs from yeast possessing complete function are found to be only composed of a single subunit, including Trm1–5p, Trm10p, Trm12p, Trm13p, and Trm44p. The modification types and positions corresponding to these TRMs are shown in Figure 1A and 1B, respectively. Table 1 summarizes the modification types, modification positions, and substrate tRNA species corresponding to various yeast single-subunit TRMs. Trm2p is known to be able to recognize all the tRNA species, so the range of substrate tRNAs of Trm2p is the widest among these single-subunit TRMs.

## 3. Overview of Yeast Single-Subunit TRMs

SAM-dependent methyltransferases can be divided into at least five independent structural classes, and the vast majority of TRMs can be assigned to classes I and IV [36]. Class I methyltransferases are characterized by the existence of a Rossmann-fold domain that can accommodate the SAM cofactor [19], while class IV methyltransferases are characterized by the folding of three β strands into a deep trefoil knot [37,38]. TrmO from *Escherichia coli* catalyzing the formation of *N*6-methylthreonylcarbamoyl adenosine (m^6^t^6^A) possesses a unique single-stranded β barrel structure, representing an additional structural class of methyltransferases [39]. The category and family affiliation of single-subunit TRMs from *Saccharomyces cerevisiae* are tabulated in Table 2. Trm3p and Trm10p are classified as class IV methyltransferases and are also members of the SPOUT RNA methyltransferase superfamily [40]. Yeast single-subunit TRMs that belong to class I methyltransferases can be further divided into two protein subfamilies according to the result of sequence homology analysis, as shown in Figure 2. 

In yeast single-subunit TRMs, only the structure of Trm10p has ever been determined, so we first carried out preliminary bioinformatic analyses of these single-subunit TRMs from *Saccharomyces cerevisiae*.

Figure 3 illustrates the comparative sequence analyses of the conserved sites of each single-subunit TRM from *Saccharomyces cerevisiae* (summarized from the annotations listed in UniProt and NCBI databases as well as the searching results by using the Pfam server). As all of these TRMs possess SAM-binding sites and catalytic domains, they can play a complete catalytic modification role in the form of a single subunit.

To search for the potential substrate tRNA-binding region of yeast single-subunit TRMs, structure predictions of Trm1–5p, Trm12p, Trm13p, and Trm44p were carried out by using the RosettaFold server [42] (robetta.bakerlab.org; accessed on 1 March 2023). Electrostatic potential analyses based on these predicted structures combined with the comparison of the annotated catalytic domain and SAM-binding sites listed above demonstrate that some of the surface patches rich with basic side chain residues marked in Figure 4 are possible regions for accommodating substrate tRNAs.

### 3.1. Yeast Trm1p

Trm1p protein encoded by the *trm1* gene from *Saccharomyces cerevisiae* methylates guanylate residue at position 26 in tRNA to form the m^2,2^G26 residue [24]. Most tRNA species can be recognized by Trm1p, including tRNA^Leu(UAA)^, tRNA^Phe(GAA)^, tRNA^Pro(UGG)^, tRNA^Ser(UGA)^, and tRNA^Leu(UAG)^. All of these tRNAs can serve as the substrate both in the cytoplasm and mitochondrion. In addition, there is a very small number of tRNA species modified by Trm1p located in the nucleus [24,43,44]. m^2,2^G26 is present in archaeal tRNAs, some eukaryotic tRNAs, and tRNAs from only one bacterial species, *Aquifex aeolicus* [45,46,47]. Position 26 is adjacent to the anticodon stem and loop, which is deduced to be crucial for the formation of m^2,2^G26 [48].

The interaction between tRNA and Trm1p does not depend on the complete tertiary structure of tRNA, and the substrate specificity determinant of this methyltransferase is located in the core of tRNA [48]. Edqvist et al. utilized two types of tRNAs, wild-type yeast tRNA^Phe^ (referred to as A) and two variants of this tRNA that respectively lack the anticodon loop and the whole anticodon stem and loop (collectively referred to as B; each variant can be correctly folded), as the substrates for the verification of the substrate specificity determinant of tRNA in Trm1p-catalyzed methylation, suggesting that G26 in both A and B can be effectively dimethylated after 30 min of incubation with Trm1p [48]. There are two conditions that need to be met for the dimethylation of G26: G-C base pairs in the D stem (such as the G10:C25 and C11:G24 base pairs) and more than four nucleotides in the variable loop [48,49]. 

Because the transcriptional product of Trm1p includes two AUGs separated by 15 codons, there are two forms of Trm1p with different N-terminal sequences. Both enzymes synthesized from the first or second AUG can be trafficked into the mitochondrion for base modification, while the base modification of tRNA in the cytoplasm is mainly carried out by the enzyme synthesized from the second AUG, indicating that the N-terminal extension is not essential for the function of Trm1p in the mitochondrion [43]. 

Through microinjection into the complete cell system of *Xenopus laevis* oocytes, it was found that the unmodified G26 in yeast tRNA^Asp^ became the monomethylated form m^2^G26, suggesting the presence of G26 monomethyltransferase in higher eukaryotic species [49]. In *Thermoplasma acidophilum*, m^2^G modification is an intermediate that is further modified to m^2,2^G [50], but no direct evidence has been found to demonstrate the presence of this intermediate relationship in the case of yeast Trm1p. 

Trmt1p protein from humans or other vertebrates is a homologue of yeast Trm1p [51,52,53] (Appendix A). Trmt1p is predicted to contain a mitochondrial targeting signal sequence as well as mitochondrial protease cleavage sites at the amino terminus, so Trmt1p is able to catalyze substrate tRNA in both the cytoplasm and mitochondria to generate m^2^G or m^2,2^G modification at position 26 [52,53]. Human Trmt1p independently catalyzes the m^2^G26 or m^2,2^G26 modification in a substrate tRNA-dependent manner. Base pairs in the D stem, such as the C11:G24 base pair and U10:A25 or G10:C25 base pair, are essential for the generation of m^2,2^G26 modification, which is also the case in other higher eukaryotic species [53]. These two G:C base pairs in the D stem, regardless of the size of the variable loop, are required for higher eukaryotic Trm1-mediated m^2,2^G26 formation, demonstrating that the substrate recognition requirements for the catalysis of m^2,2^G26 modification of yeast Trm1p are more rigorous than that of its counterpart from higher eukaryotic species [53]. Several mutations located in the methyltransferase domain of human Trmt1p have been reported to be closely related to neurological disorders [54], illustrating that methylation of G26 in tRNA is critical to the development and normal function of the neurological system.

### 3.2. Yeast Trm2p

Trm2p protein encoded by the *trm2* gene from *Saccharomyces cerevisiae* corresponds to the modifications at position 54 of all tRNAs (these following substrate tRNAs are located in the mitochondrion: tRNA^Arg(UCU)^, tRNA^Arg(ACG)^, tRNA^Gly(UCC)^, tRNA^His(GUG)^, tRNA^Ile(GAU)^, tRNA^Leu(UAA)^, tRNA^Lys(UUU)^, tRNA^Met(CAU)^, tRNA^Pro(UGG)^, tRNA^Ser(UGA)^, tRNA^Ser(GCU)^, tRNA^Thr(UAG)^, tRNA^Tyr(GUA)^), forming m^5^U54 [26,55], and the region responsible for the methyltransferase activity of Trm2p is mainly located at its C-terminal [26]. Trm2p also plays a role in DNA double-strand break repair [56]. There is evidence that Trm2p and Lhp1 proteins have a similar role in stabilizing tRNA maturation [55]. *Sup61* is a single-copy gene essential for encoding tRNA^Ser(CGA)^ in yeast [57,58]. Northern blot analysis was performed to analyze the level of tRNA^Ser(CGA)^ in wild-type, *trm2* or *sup61* single-mutant strains, and *sup61*-*trm2* double-mutant strains [55]. The results showed that there were significant differences in the amount of mature and precursor tRNA^Ser(CGA)^ between these mutant strains: the *sup61*-*trm2* double-mutant strain exhibited a decrease in the level of mature and partially processed tRNA^Ser(CGA)^ compared with that of the *sup61* mutant strain, while the level of the primary unprocessed transcript of tRNA^Ser(CGA)^ was almost identical comparing the sup61 single-mutant strain with the *sup61*-*trm2* double-mutant strain or the *trm2* mutant with the wild-type strain. These observations indicate that Trm2p does not affect the transcription of the *sup61* gene but enhances the stability of mature and partially processed tRNA^Ser(CGA)^. Although the *trm2* mutant strain showed low sensitivity to methylmethane sulfonate (MMS) [56], when *KU80*, a gene encoding a subunit of single-stranded DNA-dependent ATP-dependent helicase involved in non-homologous end-joining DNA double-strand break repair [59], was knocked out in the *trm2* mutant strain, this double-mutant strain was highly sensitive to MMS compared with that of *trm2* mutant strain [56]. In addition, the *trm2*-*rad52* double mutant also inhibited sensitivity to MMS [56]. The *rad52* gene product is necessary for DNA double-strand break repair and mating-type conversion, and mutation of this gene greatly lowers the cell mitosis level [60,61,62,63]. 

*RHO4a*, which is a homologue of *trm2 in yeast, encodes a 32kDa protein (Appendix A), showing significant sequence similarity with members of the ras family, and belongs to group IIB of ras-related proteins [64]. The C-terminal sequence is crucial for the membrane association of the RHO4a*-encoded protein [64].

### 3.3. Yeast Trm3p

In *Saccharomyces cerevisiae*, the Trm3p protein encoded by the *trm3* gene catalyzes 2′-*O* methylation at position 18 of tRNAs located in the cytoplasm (tRNA^Leu^, tRNA^His^, tRNA^Tyr^, tRNA^Ser^, and tRNA^Trp^) to form Gm18 modification [27]. The analysis of the modification pattern of tRNAs isolated from the *trm3* mutant strain and the in vitro modification experiment of the complete tRNA transcript showed that Trm3p possesses narrow substrate and modification position specificity, and the activity of Trm3p strongly depends on the conformation of mature tRNA [27]. Under different growth conditions, the destruction of the *trm3* gene will not affect the growth of *Saccharomyces cerevisiae*, indicating that Gm18 modification is not crucial for the survival of cells [27]. However, Trm3p-catalyzed Gm modification can facilitate the adaptability of protein synthesis under oxidative stress, possibly by increasing the stability of tRNA molecules under oxidative stress [34]. Moreover, Gm modification can be employed as a molecular mechanism to promote the differentiation of homologous codons and reduce the loss of codon selection in the cell partition environment (including oxidative stress), and Gm18 may interact with the D loop and T loop to contribute to the role of these two loops in maintaining the three-dimensional structure of tRNA [34].

Trm3p and human TRBP have a certain degree of homology (Appendix A). TRBP is a component of the human minimal active RNA-induced silencing complex (RISC) with Dicer and Ago2. While Dicer and Ago2 are RNases, TRBP is the double-stranded RNA-binding protein (dsRBP) that is responsible for loading small interfering RNA into the RISC. TRBP is characterized by its ability to bind the human immunodeficiency virus (HIV)-1 TAR RNA and to stimulate the expression of the HIV long terminal repeat in host cells [65,66,67,68,69]. This protein participates in the regulation of various cellular functions, including spermatogenesis, cell growth, oncogenesis, and RNA and protein binding involved in viral replication [70]. The binding of the C4 domain of TRBP to the D1 domain of Dicer is crucial for RNA interference (RNAi) activity, a natural mechanism used for gene silencing in eukaryotes [70]. When TRBP is not expressed or does not bind to Dicer, RNAi activity is significantly reduced [70]. TARBP1 and TARBP2 are isomers of TRBP. TARBP1 does not contain the zinc fingers or ribonucleoprotein-binding domain but has a leucine zipper domain in the middle of the protein for nucleic acid binding [71]. 

### 3.4. Yeast Trm4p

Trm4p protein encoded by the *trm4* gene from *Saccharomyces cerevisiae* catalyzes 2′-*O* methylation in tRNA^Leu(CAA)^ (position 34), tRNA^Phe(GAA)^ (position 40), tRNA^Tyr(GUA)^, tRNA^Ser(AGA)^ and tRNA^Ile(UAU)^ (position 48), tRNA^Phe(GAA)^ and tRNA^Asp(GUC)^ (position 49), and tRNA^His^ (positions 48, 49, and 50) to form m^5^C residue [28,29]. The substrate tRNAs of Trm4p are located in both the nucleus and cytoplasm [72]. For the substrate tRNA^Leu^, m^5^C34 exists in the wobble position of the anticodon loop, which is critical for its efficiency as a suppressor during decoding [73]. The formation of m^5^C34 and m^5^C40 strictly depends on the sequence and folding of the intron. The existence of correct base pairs and complete intron sequence is the key to the formation of m^5^C34 in tRNA^Leu^, while the m^5^C modification of positions 48 and 49 does not depend on the existence of the intron or even on the correct and complete tertiary structure of tRNA [74,75,76,77]. The m^5^C modification of tRNA^His^ is extremely sensitive to the growth conditions of yeast: Trm4p catalyzes the methylation at position 49 under normal growth conditions, while Trm4p can also catalyze the methylation of adjacent cytidines at positions 48 and 50 when Thg1 (Thg1 is a tRNA^His^ guanylyltransferase catalyzing the adding of a guanylate to the 5′ end. Loss of Thg1 corresponds to the accumulation of uncharged tRNA^His^ and leads to the arrested growth of cells) is depleted [29]. 

Yeast Pmt1 belonging to the Dnmt2 family is a homologue of Trm4p, as indicated by the sequence comparison shown in Appendix A. Dnmt2 family members are highly similar to DNA m^5^C methyltransferase in sequence [78]. Although Pmt1 contains most of the primary structural elements of a typical cytosine-specific DNA methyltransferase, it is unable to catalyze the DNA cytosine methylation due to the insertion of a Ser residue into the Pro-Cys motif generally found at the active sites of DNA m5C methyltransferases, which explains the lack of genomic methylation in *Saccharomyces cerevisiae* and *Schizosaccharomyces pombe* [78]. Notably, a Ser-Cys motif has been discovered at the active site of a methyltransferase responsible for methylations of uracil residues in tRNA, suggesting that the existence of a Ser next to a Cys in Pmt1 may not inhibit its methyltransferring activity and tRNA is probably the substrate of Pmt1 [79]. As the research further developed, it was discovered that both Pmt1 and Trm4p can mediate the activity of tRNA methylation in vivo in *Schizosaccharomyces pombe* [78]. The composition of the culture medium has a significant effect on the methylation efficiency of Pmt1: cells grown in standard complete medium were not detected for C38 methylation of tRNA^Asp^ by RNA bisulfite sequencing, while cells grown in the minimum medium showed approximately 23% C38 methylation [78]. 

### 3.5. Yeast Trm5p

Trm5p protein, encoded by the *trm5* gene from *Saccharomyces cerevisiae*, methylates the guanine nucleotide at position 37 of tRNA to form m^1^G37 [30]. The substrate tRNAs of Trm5p are distributed in the cytoplasm, mitochondrion, and nucleus [80], and Trm5p has been shown to be responsible for the m^1^G37 modification of more than 10 tRNA species, including: tRNA^Ala(IGC)^, tRNA^Leu(UAG)^, tRNA^Leu(CAA)^, tRNA^Leu(UAA)^, tRNA^Pro(UGG)^, tRNA^His(GUG)^, tRNA^Asp(GUC)^, tRNA^mtMet^, tRNA^mtPhe(GAA)^, tRNA^mtSer(UGA)^, tRNA^mtLeu(UAG)^, tRNA^mtHis(GUG)^, tRNA^mtLeu(UAA)^, and tRNA^mtPro(UGG)^ [9,25,30]. Trm5p is a class I methyltransferase which is folded in Rossmann mode for SAM binding [81]. The modification of m^1^G37 is essential for the survival of *Saccharomyces cerevisiae* [30]. The main functions of this modification are: (i) preventing +1 frameshift during ribosomal translation; (ii) regulating effective aminoacylation [82,83,84,85,86]. Taken together, this modification can reduce the error rate during the frame reading and enhance the dynamic constraint on the anticodon loop [87,88,89].

The lack of m^1^G37 modification in the *trm5* mutant strain results in growth defects [30]. In addition to m^1^G37 modification, mature tRNA^Leu(UAA)^ also has modifications at positions 32 and 34 in the anticodon loop [90,91]. Masuda et al. showed that m^1^G37 in the 3’ end of the anticodon is not directly responsible for reading of the codon but plays a role in neutralization of the differential decoding of proline codons [92]. Moreover, the loss of m^1^G37 causes the ribosome to stall on most of the relevant codons, which indicates that methylation plays a role in regulating the efficacy of the anticodon–codon pairing [93,94]. Lee et al. discovered that the efficiency and accuracy of mitochondrial protein synthesis were reduced in cells lacking m^1^G37 modification of mitochondrial tRNA, indicating that this modification plays a critical role in maintaining the normal process of mitochondrial protein synthesis [30].

TrmD, a homologue of yeast Trm5p [95] (Appendix A), is widely conserved in all bacterial species [94,96]. TrmD presents as a homodimer to bind SAM by a rare trefoil fold through hydrogen bond networks and hydrophobic interactions [97]. There are differences between TrmD and Trm5p in the recognition mode of substrate tRNA: TrmD needs an extended anticodon stem–loop and acceptor stem structure for methyltransferring activity [98], while Trm5p depends on a complete tertiary structure of tRNA [97].

Archaeal TRM5 is further divided into TRM5a, TRM5b, and TRM5c [99]. In archaea and eukaryotes, m^1^G37 of tRNA^Phe^ is modified to wyosine derivatives, which are composed of tricyclic imidazole-purine [100]. The m^1^G37 modification can make tRNA^Cys^ in some archaea, such as *Methanococcus jannaschii*, be rapidly aminoacylated [61,86] and prevent the incorrect aminoacylation by non-cognate aminoacyl tRNA synthetases while avoiding the frameshift during ribosomal translation [101,102].

### 3.6. Yeast Trm12p

Rather than the original identification as a methyltransferase, Trm12p (TYW2) encoded by the *trm12* gene from *Saccharomyces cerevisiae* is proven to be responsible for the hypermodification of position 37 of tRNA^Phe(GAA)^ in the cytoplasm [16,33,80], catalyzing the transfer of 3-amino-carboxypropyl from SAM to the C7 position of the core structure of the 4-demethylwyosine tricycle to form 7-aminocarboxypropyl-demethylwyosine (yW-72), which is finally converted to wybutosine (yW) [103]. The biosynthesis of yW in tRNA^Phe^ is a multi-enzyme process that involves five enzymes (Trm5p and TYW1–4) and may be different in various species [33,104]. The yW residue supports codon recognition by stabilizing codon–anticodon interaction during decoding in the ribosome [33].

Trm12p and VP39 have a certain degree of homology (Appendix A). VP39 is the small subunit of vaccinia virus poly(A) polymerase multi-functional polypeptide [105]. Poly(A) polymerase catalyzes the attachment of AMP to the 3′-end of mRNA [106]. Although VP39 lacks polyadenylate catalytic activity [107], it significantly accelerates the extension of either mRNA 3′-end primers with short oligo(A) tails or poly(A) primers catalyzed by E1L [108]. Moreover, VP39 can catalyze the conversion of the mRNA 5’ m^7^G (5′) pppN cap structure to m^7^G (5’) pppNm by methylation of the ribose moiety of the first transcribed nucleotide in mRNA [105]. The structure of VP39 as determined by X-ray crystallography presents a compact single domain with a typical α/β folding, with the central part of the peptide chain folded into a core structure, where the mixed seven-stranded twisted β sheet structure is surrounded by parallel α helices [105]. The residues 138–150 of VP39 are considered to be important for SAM binding and may also be important for the catalytic function of methyltransferase [105].

### 3.7. Yeast Trm13p

Trm13p encoded by the *trm13* gene from *Saccharomyces cerevisiae* is responsible for 2′-*O*-methylation at position 4 within the amino acid acceptor stem of tRNA^Gly(GCC)^, tRNA^Pro(UGG)^, and tRNA^His(GUG)^ [34,35]. The substrate tRNAs of Trm13p are located in both the cytoplasm and nucleus [80]. This modification is found to be conserved in eukaryotes. Wilkinson et al. prepared protein extracts from wild-type and *trm13* deletion strains of *Saccharomyces cerevisiae* and titrated with tRNA^Gly^, showing that 2′-*O*-methyltransferase activity was only detected in the wild-type strain, and the nucleoside composition analysis of the *trm13* deletion strain and wild-type strain by HPLC demonstrated that the *trm13* deletion strain lacked the methylation of the three tRNA species listed above [109]. 

Although the overall level of Am in the *trm13* mutant strain increased, there is a small but significant attenuation of Am level induced by hydrogen peroxide. Considering the observation that the *trm13* mutant strain exhibits a stress-sensitive phenotype, Trm13p-catalyzed Am4 modification is probably involved in responses to oxidative stress [34].

Bioinformatic analysis showed that Trm13p can be assigned to the Rossmann-fold methyltransferase superfamily (RFM) [109]. There is a zinc-binding domain possessing a CHHC zinc finger, isolated as an independent folding unit, that exists in Trm13p. This domain may serve as a tRNA recognition and binding module based on the analysis of conserved sequence characteristics [110]. It is predicted that several conserved residues of Trm13p are related to the binding of SAM: E204 can contact methionine, D243 can contact the hydroxyl group of ribose, and D282 can contact N6 of adenine [109] (the locations of these three residues are indicated in Figure 3). Moreover, the Trm13p model shows that K307, H350, E467, and several conserved cysteine residues form a pocket near the SAM-binding site, which seems to play a role in binding the target base or catalyzing the methyl transferring [109], as the corresponding cysteine residues in other RFM members (such as Trm2p and Trm4p) are involved in the methylation of cytosine C5 and uridine U5 in tRNA, generating m^5^C (by Trm2p) or m^5^U (by Trm4p) [111,112,113,114]. Nevertheless, the model suggests that some of these cysteine residues in Trm13p are unlikely to be directly involved in the methylation because these cysteine residues are far from the 2′-OH group of the target ribose. Cysteine residues (C304, C310, C343, C347, C348, and C352) and two histidine residues (H308 and H350) in Trm13p are more likely to participate in the stabilization of the protein structure via the formation of disulfide bridges and metal-coordinating bonds [109].

The human spliceosome protein U11-48K and Trm13p are homologous proteins (Appendix A). U11-48K is one of the core components of the U12-dependent spliceosome that acts on alternative splicing of pre-mRNAs [115]. U11-48K contains a CHHC zinc finger domain similar to the zinc finger domain of Trm13p [110,116]. Multiple sequence alignments constructed from U11-48K homologues reveal a highly conserved region containing four invariant Cys and His residues [116]. In the zinc finger domain of U11-48K, the Pro residue at position 5 and the Asp/Asn residue at position 7 are highly conserved, which may participate in maintaining folding stability [116]. Nuclear magnetic resonance spectroscopy and far-ultraviolet circular dichroism spectroscopy of human U11-48K protein showed that the zinc finger is located in a folding domain consisting of a β hairpin and α helix [116].

### 3.8. Yeast Trm44p

The yeast *trm44* gene encodes Trm44p, which methylates 2′-OH at position 44 of tRNA^Ser(UGA)^, tRNA^Ser(CGA)^, and tRNA^Ser(IGA)^ in the cytoplasm to form Um44 residue [20,80]. This methylated residue is located at the junction of the variable loop and anticodon stem, which is shown to play an important role in the stabilization of the structure of tRNA^Ser(CGA)^ by maximizing base interaction, thus stabilizing the endo configuration of ribose [20,117]. Another effect of Um44 is related to the aminoacylation of tRNA by seryl-tRNA synthetase, which depends on the specific recognition of the characteristics of the amino acid acceptor stem and the variable loop of tRNA by its cognate aminoacyl-tRNA synthetase [34,118]. The lack of Trm44p and tRNA-modifying enzyme Tan1 in *Saccharomyces cerevisiae* will lead to the lack of Um44 and ac^4^C12 modifications and the consequent rapid tRNA decay (RTD) of tRNA^Ser(CGA)^ and tRNA^Ser(UGA)^, resulting in growth defects [119]. During the search for genes that suppress the temperature-sensitive phenotype of *trm8*Δ-*trm4*Δ mutants, several high-copy suppressors, including TEF1 and TEF2, which each encode identical copies of elongation factor 1A (EF-1A), were found [119]. Overexpression of TEF1 or TEF2 significantly inhibits the degradation of tRNA^Val(AAC)^ in the *trm8*Δ-*trm4*Δ mutant strain undergoing RTD at 34 °C. To determine the generality of the effects of EF-1A, suppression of the growth defect of *tan1*∆-*trm44*∆ mutant strains was examined, which is also relevant to RTD of different tRNAs [119]. Because overexpression of either TEF1 or TEF2 also suppresses the temperature sensitivity of *tan1*∆-*trm44*∆ mutants and overexpression of TEF1 prevents RTD of tRNA^Ser(CGA)^ and tRNA^Ser(UGA)^ after thiolutin treatment and a shift to 36 °C, overexpression of EF-1A was deduced to generally prevent RTD. Because reduced levels of EF-1A could lead to enhanced RTD and make cells more temperature sensitive if competition between the RTD pathway and EF-1A occurs, a postulated mechanism by which RTD could be suppressed by overexpression of EF-1A is that there is competition between EF-1A and the RTD pathway for charged tRNAs in the cell, which is supported by the observations that a *tef1*∆ mutation significantly enhances the temperature sensitivity of the *tan1*∆-*trm44*∆ mutant strain, and RTD of tRNA^Ser(CGA)^ and tRNA^Ser(UGA)^ in the *tan1*∆-*trm44*∆-*tef1*∆ mutant strain is more rapid and complete compared with that of the *tan1*∆-*trm44*∆ mutant strain after thiolutin treatment and a shift to 37 °C [119].

DUF1613 is a family of eukaryotic SAM-dependent methyltransferases [120]. Trmt44p from humans is a member of the DUF1613 family and is a homologue of yeast Trm44p (Appendix A). DUF1613 proteins share the key sequence responsible for SAM binding (Motifs I–IV) (Appendix A) [120]. The conservative Asp361, Gly365, and Gly367 (corresponding to the sequence of human Trmt44p, the same in the following text) of Motif I may interact with the methionine moiety of SAM. In Motif II, Asp385, which is conserved at the β strand end of the second core, forms a hydrogen bond with the ribose hydroxyl group of SAM. Motif III lacks a conserved Asp at the end of the third core β strand. The residue of Motif IV is located between the fourth core β strand and the following α helix, possibly connected to the central part of SAM. DUF1613 protein has an absolutely conserved PxKxxxED motif (Appendix A) in the N-terminal region of the α helix preceding the first core β strand. Since the N-terminus of this α helix may be close to the methionine moiety of the SAM molecule, PxKxxXED is considered to be a Motif X responsible for SAM binding.

### 3.9. Yeast Trm10p

Trm10p has been extensively studied and only the structure of Trm10p among these single-subunit TRMs has been determined and analyzed, thus we provide a detailed description of Trm10p in terms of substrate tRNA species, modification positions, and structure–function relationships.

#### 3.9.1. Initial Identification of Yeast Trm10p

Jackman et al. initially identified tRNA methyltransferase Trm10p by a genomic method, which proved that Trm10p can specifically catalyze the formation of m^1^G9 at the junction of the amino acid acceptor stem and the D loop of tRNA both in vivo and in vitro [31,121]. Jackman et al. used the Trm10p expressed in *Escherichia coli* to modify tRNA^Gly^ in vitro, and then analyzed the hydrolysate of the product tRNA by HPLC. The result illustrated that the additional peak in the Trm10p-modified tRNA was consistent with the m^1^G nucleotide standard and had an identical spectrum to the m^1^G nucleotide standard, suggesting that the modification catalyzed by Trm10p was m^1^G [31].

#### 3.9.2. Functions of Yeast Trm10p

The m^1^G methylation of tRNA at position 9 catalyzed by Trm10p is a highly conserved modification found in eukaryotes and archaea [32]. Members of the Trm10p methyltransferase family not only catalyze the formation of tRNA m^1^G9 but also possess the activity of tRNA m^1^A9 methyltransferase. Both Trm10p and Trm5p from yeast are m^1^G methyltransferases with high specificity for their modification sites [32,122]. 

The deletion of *trm10 in Saccharomyces cerevisiae* leads to the complete loss of m^1^G9, while this mutant strain has no detectable growth defects in standard culture medium, indicating that *trm10 in Saccharomyces cerevisiae* is not essential as in the cases of many other genes responsible for tRNA modifications [31]. The substrate tRNA of Trm10p is located in both the cytoplasm and nucleus [80], and m^1^G9 modification has been discovered in more than 10 tRNA species, including tRNA^Gly^, tRNA^Trp^, tRNA^Pro^, tRNA^Val^, tRNA^Ile^, tRNA^Ala^, tRNA^Asn^, tRNA^Cys^, tRNA^Thr^, tRNA^Ser^, tRNA_i_^Met^, tRNA^Leu(CAA)^, tRNA^Arg(UCU)^, and tRNA^Arg(ICG)^ [31,32]. 

The pH rate analysis showed that for Trm10p from *Saccharomyces cerevisiae*, D210 and other conserved carboxylate-containing residues within the active site jointly established an environment required for catalysis [123]. In order to clarify the role of D210, Krishnamohan et al. carried out mutational assays of this residue to assess its impact on the methylation activity of m^1^G9 using the previously characterized tRNA^Gly^ substrate, which illustrates that mutation of this residue only moderately reduced the activity, implying that the presence of side chain carboxylate is not critical for the effective interaction with SAM during methylation [123]. Moreover, a greater combinatorial change effect than additive effect on conserved D/E residues (D100, E111, and D210. These three residues are extremely conserved in all enzymes identified so far) was observed [123]. Mutants of two and three of these residues showed similar tRNA affinity to wild-type Trm10p, which indicates that there is no obvious mutation-induced structural defect, and these three residues may play an interdependent role in maintaining the structure of the active site when binding to tRNA. The catalytic mechanism of Trm10p was deduced to be a specific base-mediated process, where the *N*1 proton is transferred to bulk solvent [123]. Therefore, the carboxylate-containing residues listed above probably indirectly participate in catalysis by promoting the organization of a solvent network to facilitate this process, and the loss of more than one of these residues is required for the disruption of this network. This proposed mechanism is in line with the observation that catalytic defects were related to the loss of more than one of these carboxylate-containing side chains.

Recently, intrinsic tRNA flexibility and enzyme-induced specific tRNA conformational change were found to be key factors for the discrimination of substrate between structurally similar tRNA species by Trm10p for methylation [124]. The conformational changes of substrate tRNA observed by using a sensitive RNA structure-probing method mainly lead to the increased reactivity in the D loop and decreased reactivity in the anticodon loop, which presumably position the target nucleotide in the binding pocket of Trm10p. A high-resolution structure is essential to reveal the details of the binding mode between Trm10p and substrate tRNA for identifying the mechanism of substrate recognition by Trm10p, which will also facilitate the understanding of the substrate recognition mechanism of other tRNA-binding proteins, as all tRNAs possess similar overall tertiary structure. 

#### 3.9.3. Structure of Yeast Trm10p

Trm10p is a member of the SPOUT superfamily, and its homologues are highly conserved in eukaryotes and archaea [123]. Through multiple sequence alignment analysis, Shao et al. found that Trm10p from *Schizosaccharomyces pombe* (SpTrm10p) and Trm10p from *Saccharomyces cerevisiae* (ScTrm10p) have similar domain composition [121]. The crystal structure of the SPOUT domain of Trm10p presents the location of the α6 helix, which can interact with the α5 helix, β4–β5 loop, and β6–α5 loop to form a hydrophobic core in this domain through hydrogen bonding and hydrophobic interactions [121]. Sequence comparison with other SPOUT domain-containing methyltransferases indicates that the α6 helix only exists in Trm10p. As the α6 helix of Trm10p will collide with the dimer interface of TrmH (a bacterial Gm18 methyltransferase) in space, the existence of the α6 helix can explain the monomer behavior of Trm10p, which is contrary to the homodimer appearance exhibited in other SPOUT enzymes [31]. Moreover, the α6 helix was deduced to be critical for the normal function of ScTrm10p in vivo, because the serine to proline mutation in the α6 helix affects the translation termination efficiency in *Saccharomyces cerevisiae*, but its exact mechanism has yet to be fully understood [125,126]. Figure 5 shows the overall structure and surface electrostatic potential of SpTrm10p, indicating the possible tRNA-binding region in Trm10p.

Moreover, yeast Trm10p contains an N-terminal extension domain, unresolved in the crystal structure, which can enhance enzymatic activity plausibly through recruiting substrate tRNA [121]. According to multiple sequence alignment analysis, this N-terminal extension domain of yeast Trm10p is not conserved in the Trm10p family and is rich with basic residues [121]. 

#### 3.9.4. Comparison of Trm10p from Other Species 

Trm10p has been discovered in eukaryotes and archaea [125]. Both eukaryotic and archaeal Trm10p exist in solution as monomers [121,126]. A prominent feature of Trm10p is the variation of substrate specificity among different homologues [31,126]. Archaea *Sulfolobus acidocaldarius* Trm10p (SaTrm10) and human Trmt10B specifically catalyze the *N*1-methylation of adenosines at position 9 of tRNAs, while yeast Trm10p and its homologue in humans, Trmt10A, are responsible for the *N*1-methylation of guanosines at position 9 of tRNAs [127,128]. In addition, Trmt10C, another Trm10p homologue in the human mitochondrion, and Trm10p from archaea *Thermococcus kodakarensis* (TkTrm10) exhibit adenosine/guanosine dual specificity [127,128].

There is a certain sequence variation between Trmt10A and Trmt10B, and the in vitro catalytic efficiency of Trmt10A is higher than that of Trmt10B [129]. Additionally, Trmt10A and Trmt10B have unique and non-overlapping substrate tRNAs and different purine specificities: Trmt10A is guanosine specific, while Trmt10B is the first adenosine-specific member discovered in the Trm10 family in eukaryotes [130]. Although knockout of Trmt10A and/or Trmt10B does not affect any major growth phenotypes in human cell lines, Trmt10A has recently been demonstrated to be associated with several serious neurological and pancreatic diseases such as microcephaly and early-onset diabetes [129], suggesting that the lack of methylation at position 9 of tRNAs presumably does not lead to severe protein synthesis deficiencies but rather to suboptimal protein synthesis, which becomes significantly harmful only in certain cell types or developmental stages or statuses in response to stimuli that require sustained high levels of protein translation. Further studies in physiologically relevant models are needed to unveil the pathogenesis corresponding to the lack of this type of tRNA modification.

Trmt10C requires the second subunit of an unusual protein ribonuclease P in human mitochondrion (mtRNase P), SDR5C1, to achieve effective pre-tRNA methylation [127,131]. mtRNase P is composed of three subunits and carries out RNA cleavage and methylation through unknown mechanisms [132]. SDR5C1 is a dehydrogenase that participates in organelle RNA processing [133]. The Trmt10C–SDR5C1 subcomplex of human mtRNase P is a bifunctional tRNA methyltransferase responsible for the formation of m^1^G9 and m^1^A9 in human mitochondrial tRNAs [131]. The structure of mtRNase P reveals the pre-tRNA substrate-binding site within TRMT10C–SDR5C1 subcomplex: the amino acid acceptor stem of pre-tRNA is located at the bottom of Trmt10C, while the anticodon loop is located adjacent to the bottom of Trmt10C and extends downward to SDR5C1 [132].

Besides D210 mentioned above, there is a second aspartate residue in SaTrm10, which is critical for catalysis [123]. Van Laer et al. found that in SaTrm10, amino acid substitution in N-terminal and C-terminal domains completely abolished its tRNA binding ability, indicating the difference in tRNA binding modes between eukaryotic and archaeal Trm10p [126]. 

TkTrm10 is a dual-specific enzyme that catalyzes the formation of m^1^A and m^1^G at the 9th position of tRNA^Asp^ species [134]. The crystal structures of the SPOUT domain of TkTrm10 complexed with the bound substrate and product show the conformational flexibility of the loop located in the active site [134].

### 3.10. Joint Function of Multiple Single-Subunit TRM-Coding Genes

The effect of the *trm* genes on the growth of yeast was further verified through mutational experiments. It was found that some of the regulation of the redox process was relieved in the *trm* (*trm3*, *trm13*, and *trm44)* mutant strains of yeast, and the corresponding *trm* mutants responsible for 2′-*O*-methylation were more sensitive to oxidative stress: after exposure to hydrogen peroxide, the Um, Cm, and Gm residues in the wild-type strain increased, but the increment of the Um, Cm, and Gm residues in the *trm* mutant strain induced by hydrogen peroxide decreased. Introduction of the coding gene of 2′-*O*-ribose methyltransferase into the corresponding deletion mutant strain can alleviate the oxidative stress-sensitive phenotype to a certain extent [27].

## 4. Conclusions and Prospects

Yeast TRM can act in the form of either a single subunit or multiple subunits, playing an important role in maintaining a large number of tRNA modifications in yeast. This review summarized the tRNA modifications catalyzed by yeast single-subunit TRMs and the corresponding substrate tRNA species. The SAM-binding sites and catalytic domains as well as possible substrate tRNA-binding regions of yeast Trm1–5p, Trm12p, Trm13p, and Trm44p have been predicted, and the structure and critical catalytic sites of Trm10p have been experimentally characterized, which provide the plausible explanation for the observation that these TRMs can play a complete catalytic modification role in the form of only a single subunit. Trm3p and Trm10p are class IV methyltransferases, while most of other yeast single-subunit TRMs mentioned above are class I methyltransferases. According to sequence similarity analysis, yeast single-subunit TRMs that are class I methyltransferases can be further divided into two subfamilies: Trm1p and Trm4p represent the first, while Trm2p, Trm5p, and Trm12p represent the second. Trm1p, Trm3p, Trm5p, and Trm10p can catalyze the modifications of guanine bases, and the corresponding modifications are m^2,2^G26, Gm, m^1^G37, and m^1^G9, respectively. Trm4p can catalyze the modification of cytosine bases, corresponding to m^5^C (positions 34, 40, 48, 49, and 50).

The methylation positions, methylation modification types, and several key residues of yeast single-subunit TRMs have been studied. Without considering the tRNA modification site, the type of methylation modification catalyzed by yeast single-subunit TRM is single (Table 1). Mutational experimentation, as one of the most important tools to effectively verify the function of the enzyme as well as its key residues, has played an important role in the functional research of yeast single-subunit TRM. For example, *trm1* mutation affects the activity of m^2,2^G methyltransferase, *trm2* mutation shows low sensitivity to MMS, and mutations of the coding genes of several single-subunit TRMs lead to growth defects. 

Although the multiple functions of yeast single-subunit TRM have been extensively studied, the corresponding structural research is still insufficient: only the structure of yeast TRM10 has ever been determined. The reliable structural information for tRNA binding of yeast single-subunit TRM is lacking, which hinders the understanding of the specific substrate tRNA recognition mechanism employed by these functionally important TRMs. In the future, with the technological progress of structure determination and structure prediction, the structural analysis of other yeast single-subunit TRMs and their complexes with substrate tRNA will be an important research direction, which will verify the binding region of tRNA of the TRMs predicted in this work and reveal more novel types of recognition patterns of tRNAs to be modified as well as the interaction details between tRNAs and these TRMs that impact the modification products.

## Figures and Tables

**Figure 1 jof-09-01030-f001:**
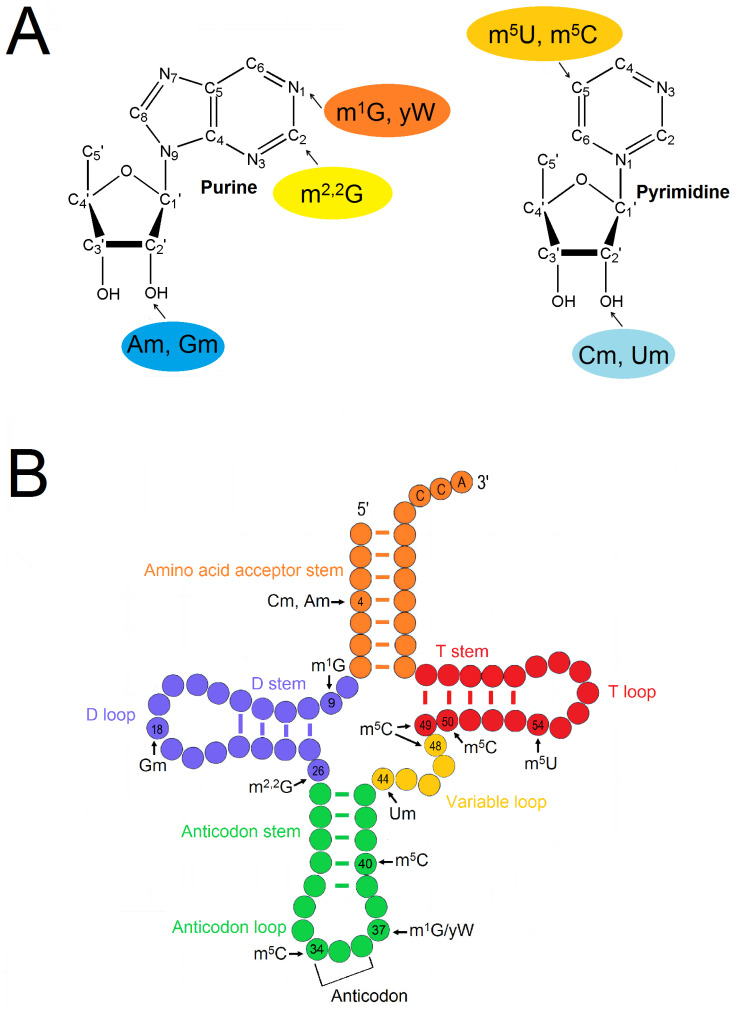
The modification types and positions of tRNAs catalyzed by single-subunit TRMs from yeast. (**A**) Methylation positions in tRNA nucleosides catalyzed by yeast single-subunit TRMs; (**B**) the secondary structure of tRNA, showing the positions modified by yeast single-subunit TRMs. All the methylation positions illustrated in the figure are not related to any particular tRNA species.

**Figure 2 jof-09-01030-f002:**
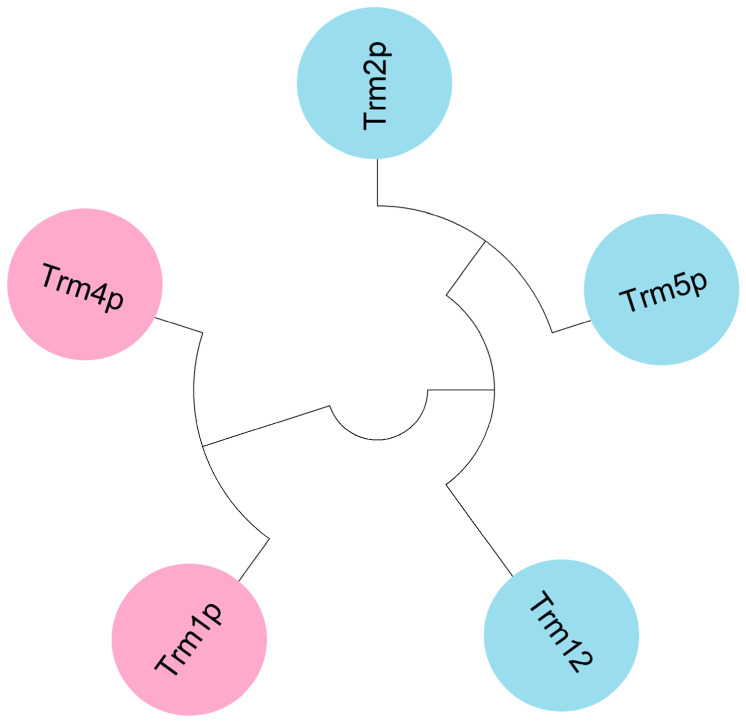
Sequence homology analysis of single-subunit TRMs belonging to class I methyltransferases from *Saccharomyces cerevisiae*. The phylogenetic tree of these TRMs generated by sequence alignment via ClustalW (using BLOSUM as the weight matrix) and then using the neighbor-joining algorithm in MEGA software suggests that they can be further divided into two protein subfamilies according to sequence similarity: Trm1p and Trm4p represent the first, while Trm2p, Trm5p, and Trm12p represent the second.

**Figure 3 jof-09-01030-f003:**
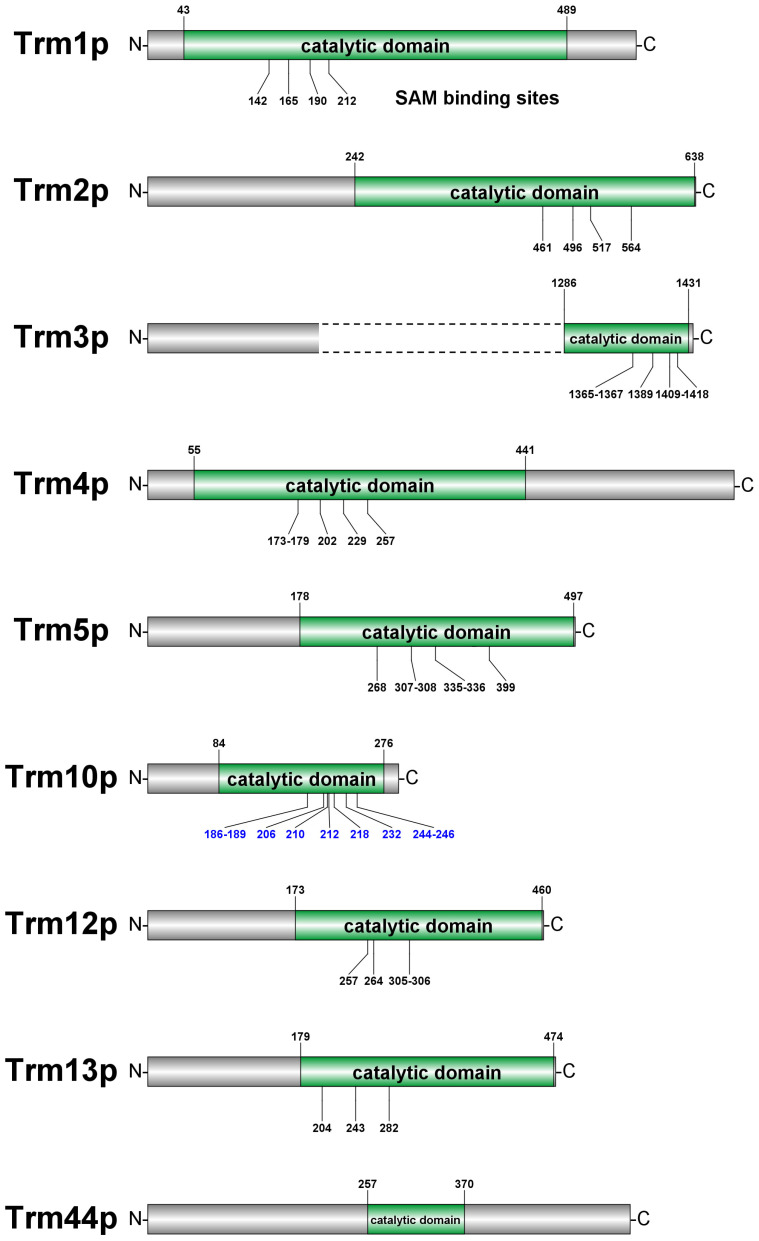
The primary structures of single-subunit TRMs from *Saccharomyces cerevisiae* indicate the catalytic domains as well as the SAM-binding sites, which are predicted or experimentally identified. The numbers above the sequence graph represent the boundaries of catalytic domains, while the numbers below represent the SAM-binding sites (the corresponding numbers in the graph of Trm10p are highlighted in blue because only these positions have been experimentally characterized). The dashed lines refer to the omitted partial sequence of Trm3p, as the sequence of Trm3p is significantly longer than those of other single-subunit TRMs. The figure was prepared using DOG software [41].

**Figure 4 jof-09-01030-f004:**
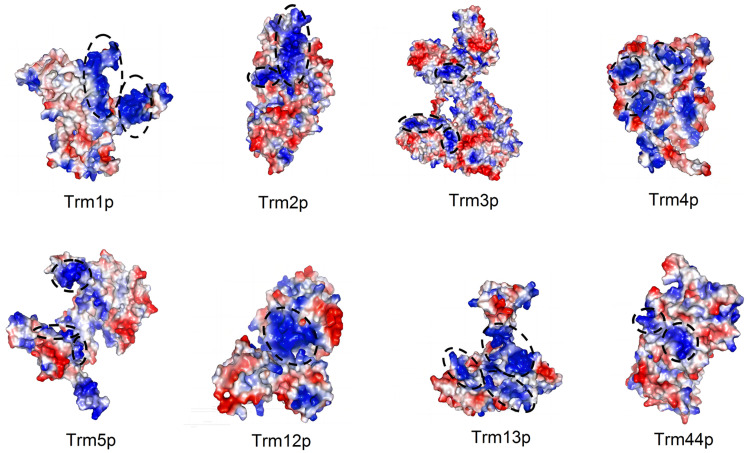
Electrostatic surface potential of the RosettaFold-predicted structures of single-subunit TRMs from *Saccharomyces cerevisiae*. The possible tRNA-binding regions predicted according to the location of predicted positive charged surface patches combined with the probable SAM-binding sites and catalytic active regions are indicated with black dashed circles.

**Figure 5 jof-09-01030-f005:**
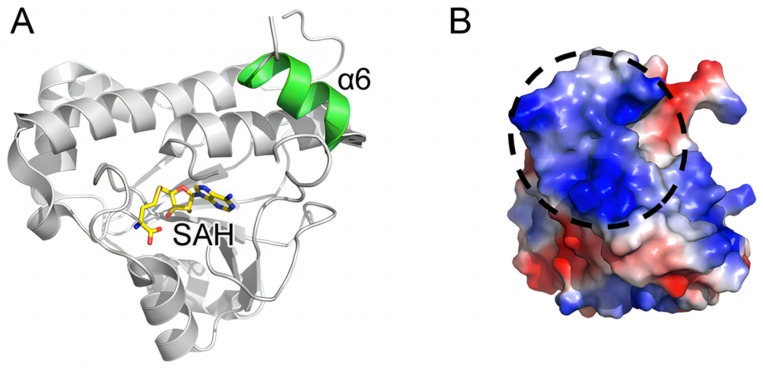
(**A**) The overall structure and (**B**) electrostatic surface potential of Trm10p from *Schizosaccharomyces pombe* (SpTrm10p). α6 helix is highlighted in green and indicated, while *S*-adenosyl-L-homocysteine (SAH) is shown as a stick model and indicated; the possible substrate tRNA-binding region is marked by the black dashed circle in the electrostatic surface potential.

**Table 1 jof-09-01030-t001:** tRNA methylations and the corresponding single-subunit TRMs from *Saccharomyces cerevisiae*.

The Type of Single-Subunit TRMs	Modification Types	Modified Positions in tRNAs	Substrate tRNA Species Recognized by Single-Subunit TRMs	References
Trm1p ^a^	m^2,2^G	26	Most tRNA species including tRNA^Leu(UAA)^, tRNA^Leu(UAG)^, tRNA^Phe(GAA)^, tRNA^Pro(UGG)^, tRNA^Ser(UGA)^	[24,25]
Trm2p	m^5^U	54	all tRNAs	[26]
Trm3p	Gm	18	Isotypes of tRNA^Leu^, tRNA^His^, tRNA^Tyr^, tRNA^Ser^ and tRNA^Trp^	[27]
Trm4p	m^5^C	34, 40, 48, 49, 50	tRNA^Leu(CAA)^ (position 34); tRNA^Phe(GAA)^ (position 40); tRNA^Tyr(GUA)^, tRNA^Ser (AGA)^ and tRNA^Ile (UAU)^ (positions 48); tRNA^Phe(GAA)^ and tRNA^Asp(GUC)^ (positions 49); tRNA^His^ (positions 48, 49, 50)	[28,29]
Trm5p	m^1^G	37	tRNA^Ala(IGC)^, tRNA^Leu(UAG)^, tRNA^Leu(CAA)^, tRNA^Leu(UAA)^, tRNA^Pro(UGG)^, tRNA^His(GUG)^, tRNA^Asp(GUC)^, tRNA^mtMet^_i_, tRNA^mtPhe(GAA)^, tRNA^mtSer(UGA)^, tRNA^mtLeu (UAG)^, tRNA^mtHis(GUG)^, tRNA^mtLeu(UAA)^, tRNA^mtPro(UGG)^	[9,25,30]
Trm10p	m^1^G	9	Isotypes of tRNA^Gly^, tRNA^Trp^, tRNA^Pro^, tRNA^Val^, tRNA^Ile^, tRNA^Ala^, tRNA^Asn^, tRNA^Cys^, tRNA^Thr^ and tRNA^Ser^; tRNA_i_^Met^, tRNA^Leu(CAA)^, tRNA^Arg(UCU)^, tRNA^Arg(ICG)^	[31,32]
Trm12p	yW (wybutosine)	37	tRNA^Phe(GAA)^	[15,33]
Trm13p	Nm (N = A or C)	4	tRNA^Gly(GCC)^ (Cm), tRNA^Pro(UGG)^ (Cm), tRNA^His(GUG)^ (Am)	[34,35]
Trm44p	Um	44	tRNA^Ser(UGA)^, tRNA^Ser(CGA)^ and tRNA^Ser(IGA)^	[20]

^a^ All of the substrate tRNA species of Trm1p are distributed in both cytoplasm and mitochondria [24,25].

**Table 2 jof-09-01030-t002:** Category and family affiliation of single-subunit TRMs from *Saccharomyces cerevisiae* summarized from the annotations listed in the UniProt database.

The Type of Single-Subunit TRM	Category	Family
Trm1p	Class I-like SAM-dependent methyltransferase superfamily	Trm1 family
Trm2p	Class I-like SAM-dependent methyltransferase superfamily	RNA m5U methyltransferase family
Trm3p	Class IV-like SAM-dependent methyltransferase superfamily	RNA methyltransferase TrmH family
Trm4p	Class I-like SAM-dependent methyltransferase superfamily	RsmB/NOP family, Trm4 subfamily
Trm5p	Class I-like SAM-dependent methyltransferase superfamily	Trm5/TYW2 family
Trm10p	Class IV-like SAM-dependent methyltransferase superfamily	Trm10 family
Trm12p	Class I-like SAM-dependent methyltransferase superfamily	Trm5/TYW2 family
Trm13p	-	Trm13 family
Trm44p	-	Trm44 family

## Data Availability

There were no new data created in this work.

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
