# Peer review of "Investigations of Single-Subunit tRNA Methyltransferases from Yeast"

_jof, 2023, doi:10.3390/jof9101030_

Round 1

Reviewer 1 Report

In this review Wang et al. summarized single subunit tRNA methyltransferase in yeast. They also performed the primary bioinformatic analyses. However, this manuscript requires many revisions. We must write a review carefully because a review is a compilation of the work of many researchers so far.

Major points

Line 53 

Since base modification is the first example of tRNA methylation, it would be better to give an example of base modification in addition to ribose methylation.

Line 58 

The reference is not properly cited. Please also check other references carefully.

Line 71-78

As an introduction, non-single-subunit tRNA methyltransferases and the methylation positions should also be provided.

Figure 1 

The figures are very hard to see, especially the yellow font. Modification should be shown lager. Fig 1A, Bases are low resolution. 

<Table 1> 

Categories are a mix of plural and singular.  >>Modification types, TRMs

The table is hard to read. I think it's easier to see if enzymes, modification types, positions, and references are aligned at the top. 

Is m2G26 intermediate of m22G26 not included in one modification type of Trm1p?

Is reference 26 in Table 1 correct? Isn’t it reference number 25?

In the table, the substrate of Trm2p is “all tRNAs”, but In Line 77 it is “most types of tRNA species”. Which is correct? 

I don't think Trm44p's references in the table are appropriate.

Notes of Table1

Isn’t it tRNALys(CUU) instead of tRNALys(UUU)? Please check it. Regarding notes a and b, it would be helpful if there were references in the text. 

It is not clear what Figure 2A means. When you show a figure, you should mention something.

Figure 2B does not show what each sequence is in the figure. 

<Figure 3>

It is easier to understand if “Trm1p, Trm2p, Trm3p...” are written directly in the figure instead of “A, B, C, D...”

The meaning of numbers under the catalytic domain in the figure should be indicated in the figure legend.

The relative positions of the numbers shown below the catalytic domain differ depending on the enzyme. Does the length of the bar correspond to the relative size of the protein?

<Figure 4>

Same as Figure 3, It is easier to understand if “Trm1p, Trm2p, Trm3p...” are written directly in the figure.

Line301-304

Is the reference correct? In this manuscript “TrmD only needs an extended anticodon stem-loop structure for methyltransferring activity” is written. Isn’t the acceptor stem also necessary? Please see Redlak et al. Biochemistry (1997).

Line 355 

When I look Fig 3H, I cannot see where E204, D243, and D282 are located because the residues are not shown.

Line 388 

“…will reduce the expression level of tRNASer” 

Does the absence of trm44 and tan1 genes reduce the expression level? I think that the tRNASer is degraded by LTD. Please check it.

Line 408 

It is necessary to explain what kind of protein DUF1613 is.

Line 410

Amino acid residues and the numbers in the main text line 410 do not coincide with those in Fig S9. Though Motif I, II, III are written in the main text, those are not shown in Fig S9. Therefore, I cannot see where the relevant motifs fall. “PxKxxXED” in Line 411 should also be clearly marked on the diagram.

Line 419-

Only Trm10p is written separately from other enzymes, the reason for which should be explained at the beginning of this section.

Line 484

It’s better to write that TrmH is bacterial Gm18 methyltransferase.

<Fiure 5>

It is easier to understand if the positions alpha 6 and AdoHcy are shown in the figure.

Minor points

Many are not italicized or superscripted. Please check out the entire text. 

“S” of S-adenosyl-L-methyonine and “N” of N6-methylthreonylcarbamoyl adenosine are italic. 

Line 347

mA4>>Am4

Line 359 

cystine>>cysteine

Line 360

“ …in the methylation of cytosine C5 in RNA, generating m5U or m5C.”

The “RNA” is “tRNA”, and m5U is uridine methylation.

Line 363

Cystine>> Cysteine residues

In the legend of Fig 5, (A) and (B) should be before each description, and S. cerevisiae is iltalic. 

Author Response

Responses to the comments and suggestions of reviewer1

We thank the reviewers for their careful read and thoughtful comments, which could significantly improve the quality of our manuscript. We have carefully taken their comments into consideration in preparing our revision and have responded to each point raised in the review. Details of the revisions made to the manuscript in response to these comments are listed below. Moreover, several additional mistakes we discovered in the previous manuscript have also been corrected. All the changes to the original manuscript have been marked in blue in the revised manuscript and supplementary material.

  1. This manuscript requires many revisions. We must write a review carefully because a review is a compilation of the work of many researchers so far.

    Sorry for the less rigorous writing in the previous manuscript. We have carefully checked through the manuscript and revised the manuscript according to the comments of reviewers. We believe that the quality of the revised manuscript has met the requirements of publication.

Major points

  1. Line 53 

Since base modification is the first example of tRNA methylation, it would be better to give an example of base modification in addition to ribose methylation.

    Thanks for this suggestion. An example of base modifications in human mitochondrial tRNAs has been added in the revised manuscript.

  1. Line 58 

The reference is not properly cited. Please also check other references carefully.

    Sorry for the incorrect citations in the previous manuscript. We have carefully checked and corrected all the citations. The citation numbers have been updated as well.

  1. Line 71-78

As an introduction, non-single-subunit tRNA methyltransferases and the methylation positions should also be provided.

    Thanks for this suggestion. Several examples of yeast multiple-subunit tRNA methyltransferases and the methylation positions catalyzed by them have been added in the revised manuscript.

  1. Figure 1 

The figures are very hard to see, especially the yellow font. Modification should be shown lager. Fig 1A, Bases are low resolution. 

    Sorry for the unclear depiction in the previous Fig. 1. We have enlarged the size of Fig. 1A, enlarged the font sizes in Fig. 1A and B, and changed the yellow font in Fig. 1B to dark-yellow. 

  1. <Table 1> 

Categories are a mix of plural and singular.  >>Modification types, TRMs

The table is hard to read. I think it's easier to see if enzymes, modification types, positions, and references are aligned at the top. 

     Sorry for the inappropriate presentation in the previous Tab. 1. All the terms in the first row of Tab. 1 have been changed to plural form, and these terms have been aligned at the top.

  1. Is m2G26 intermediate of m22G26 not included in one modification type of Trm1p?

    After surveying the literatures, we observed that in Thermoplasma acidophilum, m2G modification is an intermediate that is further modified to m2,2G, but no direct evidence has been found to demonstrate the presence of this intermediate relationship in the case of yeast Trm1p. Therefore, m2G26 cannot be considered as one modification type of Trm1p.

  1. Is reference 26 in Table 1 correct? Isn’t it reference number 25?

    Sorry for this incorrect citation in the previous Tab. 1. We have checked and corrected all the citations.

  1. In the table, the substrate of Trm2p is “all tRNAs”, but In Line 77 it is “most types of tRNA species”. Which is correct? 

    Sorry for those ambiguous descriptions in the previous manuscript. In Line 77 of the previous manuscript, “most types of tRNA species” means that the type of tRNA species recognized by Trm2p is the largest among all these single-subunit TRMs. Therefore, this is consistent with the presentation in Tab. 1 that the substrate range of Trm2p cover all tRNAs. An explanation have been added in the revised Section 2 for clarity: Trm2p is known to be able to recognize all the tRNA species, so that the range of substrate tRNAs of Trm2p is the widest among these single-subunit TRMs.

  1. I don't think Trm44p's references in the table are appropriate.

     Sorry for this incorrect citation in the previous Tab. 1. We have checked and corrected all the citations.

  1. Notes of Table1

Isn’t it tRNALys(CUU) instead of tRNALys(UUU)? Please check it. Regarding notes a and b, it would be helpful if there were references in the text. 

    Sorry for this mistake in the previous notes of Tab. 1. After carefully surveying the literature, we observed that there is currently no evidence to suggest that the m2G9 modification of tRNALys(CUU) is catalyzed by Trm10p. Therefore, this note has been deleted in the revised Tab. 1, and the citation of the other note has been added.

  1. It is not clear what Figure 2A means. When you show a figure, you should mention something. Figure 2B does not show what each sequence is in the figure. 

    Sorry for the unclear presentation in the previous legend of Fig. 2A and the missing of sequence names in Fig. 2B. Fig. 2A shows the phylogenetic tree of single-subunit TRMs from Saccharomyces cerevisiae generated by MEGA software, suggesting that they can be divided into two protein superfamilies according to sequence similarity. Fig. 2A has been cited in the second paragraph of Section 3.

  1. <Figure 3>

It is easier to understand if “Trm1p, Trm2p, Trm3p...” are written directly in the figure instead of “A, B, C, D...”.

    Thanks for this suggestion. These corrections have been performed according to this comment.

  1. The meaning of numbers under the catalytic domain in the figure should be indicated in the figure legend.

    Sorry for the missing explanation in the previous legend of Fig. 3. Numbers under the catalytic domain indicate the SAM binding sites, the corresponding description has been added in the revised legend of Fig. 3.

  1. The relative positions of the numbers shown below the catalytic domain differ depending on the enzyme. Does the length of the bar correspond to the relative size of the protein?

    Sorry for the inaccurate lengths of the bars and positions manually depicted in the previous Fig. 3. The revised figure was prepared using DOG software. The lengths of the bars and positions in the revised figure accurately correspond to the relative size and positions of these TRMs.

  1. <Figure 4>

Same as Figure 3, It is easier to understand if “Trm1p, Trm2p, Trm3p...” are written directly in the figure.

    Thanks for this suggestion. These corrections have been performed according to this comment.

  1. Line301-304

Is the reference correct? In this manuscript “TrmD only needs an extended anticodon stem-loop structure for methyltransferring activity” is written. Isn’t the acceptor stem also necessary? Please see Redlak et al. Biochemistry (1997).

    Sorry for the incorrect citation and description in the previous manuscript. After the surveying of literature according to this comment, this sentence has been corrected to "TrmD needs an extended anticodon stem-loop and acceptor stem structure for methyl transferring activity". The citations have been corrected as well.

  1. Line 355 

When I look Fig 3H, I cannot see where E204, D243, and D282 are located because the residues are not shown.

    Sorry for the unclarity in the previous Fig. 3. E204, D243, and D282 have been clearly indicated in the revised Fig. 3 and mentioned in the main text.

  1. Line 388 

“…will reduce the expression level of tRNASer” 

Does the absence of trm44 and tan1 genes reduce the expression level? I think that the tRNASer is degraded by LTD. Please check it.

    Sorry for the misunderstanding in the previous manuscript. After surveying the literature, we changed this sentence to "The lack of Trm44p and tRNA-modifying enzyme Tan1 in Saccharomyces cerevisiae will lead to the lack of ac4C12 and Um44 modifications and the consequent rapid tRNA decay (RTD) of tRNASer(CGA) and tRNASer(UGA), resulting in growth defects".

  1. Line 408 

It is necessary to explain what kind of protein DUF1613 is.

    Sorry for the unclarity in the previous description. DUF1613 is a family of eukaryotic SAM-dependent methyltransferases including Trm44 and its homologues protein Trmt44. Actually, protein DUF1613 in the previous manuscript is human Trmt44. The corresponding content in main text and supplementary material has been properly revised.

  1. Line 410

Amino acid residues and the numbers in the main text line 410 do not coincide with those in Fig S9. Though Motif I, II, III are written in the main text, those are not shown in Fig S9. Therefore, I cannot see where the relevant motifs fall. “PxKxxXED” in Line 411 should also be clearly marked on the diagram.

    Sorry for the lack of indications of these motifs in the previous Fig S9. These motifs have been properly marked in the revised Fig S9.

  1. Line 419-

Only Trm10p is written separately from other enzymes, the reason for which should be explained at the beginning of this section.

    Currently, Trm10p has been extensively studied and only the structure of Trm10p among these single-subunit TRMs has been determined and analyzed, thus we provide a detailed description of Trm10p in terms of substrate tRNA species, modification positions and structure-function relationships. This explanation has been added at the beginning of revised Section 3.9.

  1. Line 484

It’s better to write that TrmH is bacterial Gm18 methyltransferase.

    Thanks for this suggestion. The explanation of TrmH has been added in the revised manuscript.

  1. <Fiure 5>

It is easier to understand if the positions alpha 6 and AdoHcy are shown in the figure.

    Thanks for this suggestion. The positions of alpha 6 helix and AdoHcy (SAH) have been indicated in the revised Fig. 5.

Minor points

  1. Many are not italicized or superscripted. Please check out the entire text. 

“S” of S-adenosyl-L-methyonine and “N” of N6-methylthreonylcarbamoyl adenosine are italic. 

    Sorry for these mistake in the previous manuscript. The corresponding corrections have been performed according to this comment.

  1. Line 347

mA4>>Am4

    Sorry for this incorrect presentation in the previous manuscript. This correction has been performed.

  1. Line 359 

cystine>>cysteine

    Sorry for this mistake in the previous manuscript. The corresponding correction has been performed.

  1. Line 360

“ …in the methylation of cytosine C5 in RNA, generating m5U or m5C.”

The “RNA” is “tRNA”, and m5U is uridine methylation.

    Sorry for the improper presentation in the previous manuscript. This sentence has been changed to "…in the methylation of cytosine C5 and uridine U5 in tRNA, generating m5C (by Trm2p) or m5U (by Trm4p)" for clarity.

  1. Line 363

Cystine>> Cysteine residues

    Sorry for this mistake in the previous manuscript. The corresponding correction has been performed.

  1. In the legend of Fig 5, (A) and (B) should be before each description, and S. cerevisiae is iltalic. 

    Sorry for the inappropriate presentation in the previous legend of Fig 5. The corresponding corrections have been performed.

Reviewer 2 Report

Wang et al. provide a literature review of a family of tRNA methyltransferases, focusing on those that appear to function as single subunits. The authors also provide additional, informative sequence comparisons and protein structural analyses as well. Overall, the topic of this manuscript is well-focused and addresses an important area of tRNA methylation biology. The structural discussion was quite helpful. However, the protein similarity/homolog discussion came across as a bit confusing or questionable. Comments can be found below:

1.     Some word labels in Figure 1 are blurry, especially Figure 1A. Authors should modify to make all words appear clear.

2.     On line 90, the authors state “tRNA methyltransferases can be divided into three categories.” They continue to describe class I and IV enzymes. First, authors should clarify what they mean by categories. Second, authors should at least briefly explain the three categories.

3.     As each methyltransferase discussed act as single subunits, it would be helpful to explain (if known) how each recognizes, binds, modifies, and releases their tRNA substrates. This information seems to be described for some enzymes but not others.

4.     The authors should indicate known or predicted cellular localization of each methyltransferase enzyme. In the background, the authors mentioned that they could be in nucleus or cytoplasm, which affects their functions. Thus, the localization information would be very interesting to know.

5.     The authors discuss homologs of each enzyme and also show alignment data as Supplementary material.

a.     First, in the Supplementary material, authors should indicate the overall protein sequence identity and similarity that you typically get from a sequence alignment. This will help the read assess overall and local similarities.

b.     Second, some of the homolog relationships mentioned appear unsupported by the literature. For example, the authors mention La protein as a homolog of Trm1. Also Trm3 and human TRBP as being homologs. However, overall protein sequence identity and similarities are well-below 10%. Also, from the Supplementary material, this reviewer do not see significantly conserved protein domains (only randomly dispersed protein amino acids). The authors must either reference literature that clearly supports the homology-relationship or the authors must clearly define how they call homologs.

6.     In Figure 3, the authors label “SAM binding sites”. However the figure legend suggest that at least some sites are predicted rather than experimentally verified. It would be helpful to indicate which sites are known versus predicted.

There are several grammatical errors throughout the manuscript. Authors should use a free service such as Grammarly to help improve the writing.

Author Response

Responses to the comments and suggestions of reviewer2

We thank the reviewers for their careful read and thoughtful comments, which could significantly improve the quality of our manuscript. We have carefully taken their comments into consideration in preparing our revision and have responded to each point raised in the previous review. Details of the revisions made to the manuscript in response to these comments are listed below. Moreover, several additional mistakes we discovered in the previous manuscript have also been corrected. All the changes to the original manuscript have been marked in blue in the revised manuscript and supplementary material.

  1. The protein similarity/homolog discussion came across as a bit confusing or questionable.

    Sorry for the inaccurate alignment and unclear definition of homologous proteins in the previous manuscript. We realigned the sequences by using ClustalW, ensuring the homology of most of these proteins. Due to the low sequence similarity between La protein and Trm1p, the homologous protein of Trm1p was substituted by human Trmt1. The corresponding content in the main text and supplementary material has been properly revised.

  1. Some word labels in Figure 1 are blurry, especially Figure 1A. Authors should modify to make all words appear clear.

    Sorry for the unclear depiction in the previous Fig. 1. We have enlarged the size of Fig. 1A and enlarged the font sizes in Fig. 1A and B as well. 

  1. On line 90, the authors state “tRNA methyltransferases can be divided into three categories.” They continue to describe class I and IV enzymes. First, authors should clarify what they mean by categories. Second, authors should at least briefly explain the three categories.

    Sorry for the ambiguous description of classification of tRNA methyltransferases in the previous manuscript. After surveying the literature, we changed this sentence to "SAM-dependent methyltransferases can be divided into at least 5 independent structural classes, and the vast majority of TRMs can be assigned to class I and IV".

  1. As each methyltransferase discussed act as single subunits, it would be helpful to explain (if known) how each recognizes, binds, modifies, and releases their tRNA substrates. This information seems to be described for some enzymes but not others.

    Due to the lack of reliable structural information on tRNA-binding, recognition and catalytic mechanism of most of yeast single-subunit TRMs is yet to be revealed. Therefore, only the corresponding detailed mechanism of Trm10p, structure and critical residues of which have been experimentally characterized, is realized. Additionally, only partial knowledge of tRNA recognition pattern of other single-subunit TRMs, such as Trm1p, Trm2p, Trm4p and Trm5p, has been observed. All of the corresponding descriptions have been included in the manuscript.

  1. The authors should indicate known or predicted cellular localization of each methyltransferase enzyme. In the background, the authors mentioned that they could be in nucleus or cytoplasm, which affects their functions. Thus, the localization information would be very interesting to know.

    Thanks for this suggestion. All the experimentally verified or predicted cellular localizations of these TRMs have properly been indicated in the revised manuscript.

  1. In the Supplementary material, authors should indicate the overall protein sequence identity and similarity that you typically get from a sequence alignment. This will help the read assess overall and local similarities.

    Sorry for the lack of description of a quantitative index of sequence comparison in the previous supplementary material. We realigned the sequences by using ClustalW, and calculated the overall sequence identities and similarities, which have been explicitly listed in the legends of revised supplementary figures. The overall similarities between each yeast single-subunit TRM and its homologous protein are close to or exceed 30%, ensuring the homology of all of these proteins.

  1. Some of the homolog relationships mentioned appear unsupported by the literature. For example, the authors mention La protein as a homolog of Trm1. Also Trm3 and human TRBP as being homologs. However, overall protein sequence identity and similarities are well-below 10%. Also, from the Supplementary material, this reviewer do not see significantly conserved protein domains (only randomly dispersed protein amino acids). The authors must either reference literature that clearly supports the homology-relationship or the authors must clearly define how they call homologs.

    Sorry for the inaccurate alignment and unclear definition of homologous proteins in the previous manuscript. We realigned the sequences by using ClustalW, ensuring the homology of all of these proteins, including Trm3p and human TRBP, except for La protein and Trm1p. Due to the low sequence similarity between La protein and Trm1p, the homologous protein of Trm1p was substituted by human Trmt1. The corresponding content in the main text and supplementary material has been properly revised.

  1. In Figure 3, the authors label “SAM binding sites”. However the figure legend suggest that at least some sites are predicted rather than experimentally verified. It would be helpful to indicate which sites are known versus predicted.

    Sorry for the missing of the indications of predicted and experimentally verified sites in the previous Fig. 3. Currently, only SAM binding sites in Trm10p are experimentally characterized, which are highlighted in blue in the revised Fig. 3 and clearly described in the revised legend of Fig. 3.

  1. There are several grammatical errors throughout the manuscript. Authors should use a free service such as Grammarly to help improve the writing.

    Sorry for the grammatical errors in the previous manuscript. We have corrected all the errors after checking the manuscript via Grammarly server.

Round 2

Reviewer 1 Report

Fig 1A

The numbering of purine is incorrect. N7 does not bind to C1’ of ribose.

Hydroxy group is needed at C3’ of ribose.

In the figure legend.  “(A) Methylation positions in tRNA nucleotides catalyzed…”

The figures are not nucleotides, but nucleosides.

Fig 1B

3’ end of tRNA is CCA, not ACC.

Fig 2A, B

These are major points.

Only TRM3 and TRM10 are members of the SPOUT superfamily. I do not understand why TRM3 and TRM4 are close in the tree.

It would be fine if you were comparing Rossman fold proteins.

In addition, the conserved amino acids of TRM3 and TRM10 of SPOUT differ from those of the Rossman fold protein, so the sequence alignment is nonsense.

Please refer to the review “Hori Biomolecules 2017".

In the sequence alignment,

what protein does the amino acid number correspond to?

line 169 

m2,2G26 is present in archaea tRNAs and some eukaryotic tRNAs.

Among bacteria, only Aquifex aeolicus has Trm1 (Awai JBC 2009)

176, 177 referred to A or referred to B

 I could not understand the A and B.

Author Response

We thank the reviewers for their careful read and thoughtful comments, which could significantly improve the quality of our manuscript. We have carefully taken their comments into consideration in preparing our revision and have responded to each point raised in the review. Details of the revisions made to the manuscript in response to these comments are listed below. All the changes to the previous manuscript have been marked in blue in the revised manuscript.

  1. Fig 1A

The numbering of purine is incorrect. N7 does not bind to C1’ of ribose.

Hydroxy group is needed at C3’ of ribose.

    Sorry for the mistakes and missing groups in the previous Fig. 1A. We have corrected the numbering of purine and added the hydroxy groups to the C3' positions of ribose moieties.

  1. In the figure legend.  “(A) Methylation positions in tRNA nucleotides catalyzed…”

The figures are not nucleotides, but nucleosides.

    Sorry for the mistakes in the previous figure legend. The correction has been performed according to this comment.

  1. Fig 1B

3’ end of tRNA is CCA, not ACC.

    Sorry for this mistake in the previous Fig. 1B. The correction has been performed according to this comment.

  1. Fig 2A, B

These are major points.

Only TRM3 and TRM10 are members of the SPOUT superfamily. I do not understand why TRM3 and TRM4 are close in the tree.

It would be fine if you were comparing Rossman fold proteins.

In addition, the conserved amino acids of TRM3 and TRM10 of SPOUT differ from those of the Rossman fold protein, so the sequence alignment is nonsense.

Please refer to the review “Hori Biomolecules 2017".

    Sorry for the inappropriate assignment in the previous phylogenetic tree and the nonsense sequence alignment, and thanks for this suggestion. We have realized that the sequence homology analysis of all the yeast single-subunit TRMs cannot draw valid conclusion. Therefore, we realigned the sequences of the yeast single-subunit TRMs that belong to class I methyltransferase family containing Rossmann-fold domain via ClustalW (using BLOSUM as the weight matrix), and then regenerated the phylogenetic tree by using neighbor-joining algorithm in MEGA software, illustrating that these TRMs can be divided into two protein subfamilies according to sequence similarity: Trm1p and Trm4p represent the first, while Trm2p, Trm5p and Trm12p represent the second. Due to the low overall sequence similarity among these TRMs, Fig. 2B and Fig. S1 in the previous manuscript and supplementary material have been removed. The corresponding descriptions in the main-text and figure legend have been properly revised.

  1. In the sequence alignment,

what protein does the amino acid number correspond to?

    Sorry for the missing of numbering in the previous figure legend of Fig. 2B. The sequence numbering in the previous Fig. 2B corresponds to that of Trm3p. Due to the low overall sequence similarity amomg these TRMs, Fig. 2B in the previous manuscript has been removed.

  1. line 169 

m2,2G26 is present in archaea tRNAs and some eukaryotic tRNAs.

Among bacteria, only Aquifex aeolicus has Trm1 (Awai JBC 2009)

     Thanks for this comment. The corresponding sentence has been corrected to "m2,2G26 is present in archaeal tRNAs, some eukaryotic tRNAs and tRNAs from only one bacteria species Aquifex aeolicus". The citation of T Awai et al. JBC 2009 has also been added.

  1. 176, 177 referred to A or referred to B

 I could not understand the A and B.

    Sorry for the unclear descriptions in the previous manuscript. A stands for wild-type yeast tRNAPhe, while B stands for two variants of yeast tRNAPhe that lack the anticodon loop and the whole anticodon stem and loop, respectively. Each variant can be correctly folded. These two types of tRNA substrate were employed to verify the substrate specificity determinant of tRNA in Trm1p-catalyzed methylation. The corresponding sentence has been changed to "Edqvist et al. utilized two types of tRNAs, wild-type yeast tRNAPhe (referred to as A) and two variants of this tRNA that respectively lack the anticodon loop and the whole anticodon stem and loop (collectively referred to as B; each variant can be correctly folded), as the substrates for the verification of substrate specificity determinant of tRNA in Trm1p-catalyzed methylation, suggesting that G26 in both A and B can be effectively dimethylated after 30 min of incubation with Trm1p." for clarity.

Reviewer 2 Report

none.

Author Response

none.

Round 3

Reviewer 1 Report

_

Author Response

None.